# Feature Interaction-Based Face De-Morphing Factor Prediction for Restoring Accomplice’s Facial Image

**DOI:** 10.3390/s24175504

**Published:** 2024-08-25

**Authors:** Juan Cai, Qiangqiang Duan, Min Long, Le-Bing Zhang, Xiangling Ding

**Affiliations:** 1School of Physics Electronics and Intelligent Manufacturing, Huaihua University, Huaihua 418000, China; hhxycj@hhtc.edu.cn; 2School of Computer and Communication Engineering, Changsha University of Science and Technology, Changsha 410114, China; duanqq@stu.csust.edu.cn; 3School of Electronics and Communication Engineering, Guangzhou University, Guangzhou 511370, China; 4School of Computer and Artificial Intelligence, Huaihua University, Huaihua 418000, China; zhanglebing@hhtc.edu.cn; 5School of Computer Science and Engineering, Hunan University of Science and Technology, Xiangtan 411201, China; xianglingding@hnust.edu.cn

**Keywords:** face de-morphing, face morphing attack, face recognition, feature interaction

## Abstract

Face morphing attacks disrupt the essential correlation between a face image and its identity information, posing a significant challenge to face recognition systems. Despite advancements in face morphing attack detection methods, these techniques cannot reconstruct the face images of accomplices. Existing deep learning-based face de-morphing techniques have mainly focused on identity disentanglement, overlooking the morphing factors inherent in the morphed images. This paper introduces a novel face de-morphing method to restore the identity information of accomplices by predicting the corresponding de-morphing factor. To obtain reasonable de-morphing factors, a channel-wise attention mechanism is employed to perform feature interaction, and the correlation between the morphed image and the real-time captured reference image is integrated to promote the prediction of the de-morphing factor. Furthermore, the identity information of the accomplice is restored by mapping the morphed and reference images into the StyleGAN latent space and performing inverse linear interpolation using the predicted de-morphing factor. Experimental results demonstrate the superiority of this method in restoring accomplice facial images, achieving improved restoration accuracy and image quality compared to existing techniques.

## 1. Introduction

Face recognition technology rapidly and accurately identifies individuals by analyzing facial features such as geometry and texture. This technology has found widespread applications in areas like identity verification and public security. However, its extensive use has brought to light significant security concerns, particularly regarding face morphing attacks. A face morphing attack blends multiple face images into a new image, which is then used for face recognition. Since the morphed image contains the facial features of multiple individuals, it can lead to false identity matches in face recognition systems (FRSs), thereby bypassing the authentication step. Ferrara et al. [1] demonstrated that a morphed face image can successfully match with multiple individuals in FRSs. If such an morphed face image is registered as an identity document, it poses a serious threat to the fundamental principle that an identity document should uniquely correspond to its holder, potentially facilitating illegal activities.

Morphed faces can be created using landmark-based methods, which involve linear interpolation and blending of texture features at the image level [1,2,3]. However, these methods often produce artifacts when pixels are not aligned during the interpolation and blending processes. Alternatively, feature-level face morphing methods interpolate facial features and decode them using deep learning architectures [4,5,6,7,8]. Both landmark-based and deep learning-based face morphing attacks pose significant security threats to commercial FRSs [9,10,11].

To counter these threats, morphing attack detection (MAD) has become a crucial component in resisting face morphing attacks. MAD can be generally categorized into single image-based MAD (S-MAD) [12,13,14,15,16,17,18,19,20] and differential image-based MAD (D-MAD) [21,22,23,24]. S-MAD involves extracting features from a single input image to determine if it has been morphed, while D-MAD compares feature differences between a real-time captured image and an input image to detect morphing.

While MAD methods improve FRS security, they do not restore the contributors’ face of the morphed image. Restoring the identity of a contributor is essential for forensic processes. To address this, Ferrara et al. [25] proposed a face de-morphing method to reconstruct the accomplice’s face image. However, this method relies on prior knowledge, such as morphing operation and morphing factor. When the de-morphing factor is inconsistent with the morphing factor, the effectiveness of this method is compromised. Subsequently, FD-GAN [26] employs a dual symmetric network and two-level loss to restore the accomplice’s identity without requiring morphing priors. Recently, a method to restore the identity information of contributors based on a single morphed face image [27] has been proposed. Long et al. [28] performed a face de-morphing task using a pre-trained diffusion autoencoder.

This paper presents a novel face de-morphing method that significantly differs from previous approaches: (1) The method for obtaining the accomplice’s identity features is distinct. Unlike existing deep learning-based de-morphing methods that directly rely on identity disentanglement to extract the accomplice’s identity features, the proposed approach first utilizes a cross-attention mechanism in the latent space to calculate the de-morphing factor. This factor is then used to obtain the latent identity features of the accomplice, reducing the complexity of direct identity disentanglement and improving the accuracy of identity recovery. (2) The method of generating the accomplice’s facial image is different. Considering the limited quality of existing face images restored by simple generators or landmark-based morphing inversion methods, this paper employs a pre-trained StyleGAN inversion model to generate facial images. This scheme helps produce high-resolution and high-quality accomplice’s facial images. The main contributions are as follows:A novel face de-morphing method is proposed, which significantly diverges from direct identity disentanglement approaches. This method leverages a pre-trained StyleGAN inversion model to embed facial images into a latent space and employs the predicted de-morphing factors to perform inverse linear interpolation within the latent space, thereby obtaining the identity features of accomplices to generate high-quality facial images.A feature interaction-based de-morphing factor prediction network is proposed, employing channel-wise attention mechanisms to effectively integrate features from the morphed face and the real-time captured face. This approach enables the prediction of the de-morphing factor by exploring feature correlations.The experimental results demonstrate that the proposed method can effectively restore the accomplice’s face images. Its image quality and restoration accuracy outperform those of existing face de-morphing methods.

The rest of this paper is organized as follows: Section 2 presents the related work, Section 3 elaborates on the proposed method, Section 4 presents the experimental results, and Section 5 concludes this paper.

## 2. Related Work

### 2.1. Generation of Morphed Face Images

Existing face morphing methods can be broadly categorized into two main types based on the morphing process: landmark-based generation and deep learning-based generation.

#### 2.1.1. Landmark-Based Generation

Landmark-based methods primarily generate morphed faces by interpolating facial landmarks and blending facial texture features. Ferrara et al. [1] proposed a face morphing method that, while posing a threat to commercial FRSs, is time-consuming due to the required manual intervention. Subsequently, Markrushin et al. [2] proposed complete and splicing morphing techniques for automatic morphed face image generation. Qin et al. [3] further refined these methods by introducing local face morphing, focusing on key facial regions such as the eyes, nose, and mouth.

With advancements in face morphing techniques, numerous open-source tools have emerged, including OpenCV [29] and FaceMorpher [30]. OpenCV utilizes the Dlib [31] library to detect facial landmarks and constructs Delaunay triangulation for affine transformation, enabling pixel-level morphing based on the morphing factor. FaceMorpher uses STASM [32] for landmark detection and performs similar operations to OpenCV. However, open-source tools often require significant post-processing to eliminate artifacts. In contrast, commercial tools like FantaMorph [33] incorporate advanced post-processing techniques, allowing for the batch generation of high-quality morphed face images.

#### 2.1.2. Deep Learning-Based Generation

The deep learning-based method uses an encoder to obtain the latent features of two face images for linear interpolation, and the interpolated features are decoded to generate the morphed face. The pioneering approach, MorGAN [4], maps images into the latent space of a Generative Adversarial Network (GAN) and interpolates latent features to generate morphed faces. However, MorGAN typically produces low-resolution images. To address this, Venkatesh et al. [5] employed StyleGAN [34,35] to generate high-resolution morphed faces, though with low structural similarity to the contributors’ facial structures. MIPGAN [6] improved upon this by leveraging the identity prior from pre-trained StyleGAN models, optimizing interpolated features to create more identity-protective images. Recently, methods utilizing pre-trained diffusion autoencoders have emerged [7,8], which interpolate semantic and random latent representations before decoding to generate morphed face images.

### 2.2. Face Morphing Attack Detection

Existing MADs can be categorized into two main types based on whether a reference image is utilized: single image-based morphing attack detection (S-MAD) and differential image-based morphing attack detection (D-MAD).

#### 2.2.1. Single Image-Based Morphing Attack Detection

S-MAD aims to determine whether a single image is a morphed face. Raghavendra et al. [12] leveraged texture differences between morphed and real faces using binarized statistical image features (BSIFs) to detect morphed faces. Makrushin et al. [2] proposed using JPEG compression artifacts, exploiting quantized discrete cosine transform to extract Benford features for detection. Zhang et al. [13] utilized sensor pattern noise (SPN) differences in a specific frequency domain for effective detection. With the advent of deep learning, various models have been employed for MAD. Raghavendra et al. [14] used transfer learning with pre-trained AlexNet and VGG19, fine-tuning them for MAD. Zhang et al. [15] developed a multi-scale attention convolutional neural network, achieving effective detection by using an attention recursive architecture to locate artifact regions. Soleymani et al. [16] introduced a disentanglement network trained on triples of face images, focusing on landmarks and facial appearance. Damer et al. [17] improved detection performance on re-digitized morphed images by adopting pixel-level supervision. Qin et al. [18] employed fine-grained feature-level supervision to enhance detection of local morphing regions. Kashiani et al. [19] proposed inter-domain style mixup and self-morphing augmentations to improve model generalization.

#### 2.2.2. Differential Image-Based Morphing Attack Detection

D-MAD assesses the difference between a suspicious image and a real-time captured image by an FRS. This technique is commonly employed in border control, where real-time images are compared with passport images. Feature difference-based D-MAD methods initially utilized landmarks [20] and texture information [21] for detection. Scherhag et al. [22] applied deep face representation to D-MAD, achieving high detection performance. Chaudhary et al. [23] used 2D discrete wavelet transform for decomposition and neural network training using wavelet sub-bands. Face de-morphing was introduced by Ferrara et al. [25] to restore accomplices’ facial images, using an inverse landmark-based method. Peng et al. [26] utilized a generative adversarial network for face de-morphing without prior knowledge, employing a dual symmetric network and dual-loss architecture. Ortega-Delcampo et al. [24] and Banerjee et al. [36] explored convolutional neural networks and conditional generative adversarial networks for face de-morphing, respectively. Long et al. [37] proposed a method based on multi-scale feature interaction to predict the de-morphing factor and then used the inverse operation based on landmark-based face morphing to restore the assistant’s facial image. More recently, methods have been proposed to restore the identities of two contributors from a single morphed face image [27]. Long et al. [28] employed a pre-trained diffusion autoencoder, mapping faces to semantic and random latent spaces and designing a dual-branch feature separation network for semantic latent feature extraction.

To improve image quality and restoration accuracy, the proposed approach utilizes a pre-trained StyleGAN facial inversion as the backbone network, coupled with a feature interaction-based de-morphing factor prediction network. The de-morphing factor is then utilized to restore latent features of accomplices within the latent space of StyleGAN.

## 3. Proposed Method

### 3.1. Motivation

Considering that both landmark-based and deep learning-based morphing generation methods use linear interpolation of contributors’ identity information based on morphing factors to construct morphed facial identities, identity restoration can be achieved by obtaining the de-morphing factor corresponding to the morphed facial image and performing inverse linear interpolation. Therefore, this paper proposes a feature interaction-based de-morphing factor prediction network, which leverages cross-attention mechanisms to integrate the correlation between morphed facial images and images captured by FRSs to facilitate de-morphing factor learning. Additionally, StyleGAN inversion [38,39] can map real facial images into latent space, enabling high-quality facial image reconstruction and allowing for facial editing through latent feature manipulation. Thus, this paper maps facial images into the StyleGAN latent space and utilizes de-morphing factors for inverse linear interpolation to restore the latent features of accomplices, thereby achieving face de-morphing.

### 3.2. Overall Framework

The proposed method primarily utilizes channel-wise attention mechanisms for interaction between the features of the morphed facial image and the reference facial image, aiming to predict the de-morphing factor corresponding to the morphed facial image and thereby recover the accomplice’s facial image. Figure 1 illustrates the overall architecture of the proposed method, which incorporates a pre-trained StyleGAN-based facial inversion model as the backbone network. Additionally, it constructs a feature interaction-based de-morphing factor prediction network (DMFP) to learn the de-morphing factors corresponding to the morphed facial images.

Firstly, given the morphed facial image $I_{ab}$ and the reference facial image $I_{a}$, the encoders *E* maps these input facial images into the latent space $W^{+}$, obtaining their corresponding latent codes $W_{ab}^{+}$ and $W_{a}^{+}$. Simultaneously, this method selects the feature maps $f_{ab}$ and $f_{a}$ from the top layer output of the encoders as the input to the feature interaction-based de-morphing factor prediction network. Next, the feature interaction-based de-morphing factor prediction network establishes feature interaction between the features $f_{ab}$ and $f_{a}$ using channel-wise attention mechanisms to predict the de-morphing factor α corresponding to the morphed facial image. Then, utilizing the predicted de-morphing factor, inverse linear interpolation is performed in the latent space to obtain the latent code $W_{b}^{+}$ of the accomplice facial image from the latent code $W_{ab}^{+}$ of the morphed facial image. The calculation process of inverse linear interpolation can be expressed as:(1)$$W_{b}^{+}=\frac{W_{ab}^{+}-{\alpha W}_{a}^{+}}{1-\alpha }.$$

Finally, the obtained 18,512-dimensional latent vectors are inputted into the corresponding input layer of the pre-trained StyleGAN to generate the facial image of the accomplice.

### 3.3. Feature Interaction-Based De-Morphing Factor Prediction Network

Both landmark-based and deep learning-based morphing generation methods perform linear interpolation of contributors’ identity information based on the morphing factor to construct the morphed facial identity information. Therefore, it is possible to obtain the accomplice’s identity features by learning the de-morphing factor α for inverse linear interpolation and then restore the facial image of the accomplice. To achieve this goal, this method constructs a feature interaction-based de-morphing factor prediction network to learn the de-morphing factor corresponding to the morphed facial image, integrating the feature information of the morphed and the reference facial image. The structure details of the feature interaction-based de-morphing factor prediction network are shown in Figure 2. It mainly consists of 1×1 convolution, depthwise separable convolution, and an attention mechanism. In the attention mechanism, this method uses the features $f_{a}$ of the reference facial image as the query and the features $f_{ab}$ of the morphed facial image as the key and value. To fully utilize the channel information of features, this method adopts a channel-wise attention mechanism [40] to establish information interaction between the features of the morphed and the reference facial image. Compared to traditional attention mechanisms that focus on spatial relationships, channel-wise attention mechanisms can effectively reduce model complexity.

Specifically, as depicted in Figure 2, the feature interaction-based de-morphing factor prediction model first transforms the features $f_{a}$ of the reference facial image into a query matrix (*Q*) using a 1×1 convolution followed by a 3×3 depth-wise separable convolution. Similarly, the key matrix (*K*) and value matrix (*V*) are obtained in the same manner based on the features $f_{ab}$ of the morphed facial image. Given the *Q*, *K*, and *V* matrices, channel feature interaction is achieved through matrix multiplication, where the size of the attention map is $R^{C\times C}$, with *C* representing the channel dimension. The specific calculation process of the channel-wise attention mechanism is as follows:(2)Attention(Q,K,V)=V·Sofatmax(K·Q).

Next, following the channel-wise attention mechanism, the architecture employs a feedforward network to further aggregate the features outputted by the attention module. The feedforward network primarily comprises two branches of depth-wise separable convolutions, which extract spatial and channel information further using depth-wise separable convolutions. Subsequently, the features from the two branches are integrated through element-wise multiplication. Finally, the de-morphing factor α is obtained using an average pooling layer and a fully connected layer. The specific calculation process of the de-morphing factor α can be summarized as:(3)$$Att=ChanAtten(f_{ab},f_{a}),$$
(4)Out=FN(LN(Att))+Att,
(5)α=FC(AvgPool(Out)).
where ChanAtten represents the channel-wise attention module; FN is the feedforward network; LN is layer normalization; FC and AvgPool denote the fully connected layer and the average pooling layer, respectively.

## 4. Experiments

### 4.1. Datasets

To evaluate the effectiveness of the proposed method, experiments were conducted on the HNU-FM [41] dataset and MIPGAN dataset from SYN-MAD [42]. HNU-FM adopts a landmark-based method, and all morphed facial images in the dataset have been verified by Face++ [43]. It consists of four sub-datasets: Protocol I, Protocol II, Protocol III, and Protocol IV. For our experiment, Protocol I and Protocol III datasets are selected, each containing 100 subjects (60 males and 40 females). In Protocol I, the pixel morphing factor is set to 0.5. The training set comprises 1121 morphed facial images and 1121 real facial images, the validation set consists of 564 morphed facial images and 299 real facial images, and the test set comprises 566 morphed facial images and 296 real facial images. For Protocol III, the pixel morphing factor varies between 0.1 and 0.9. It is divided into training set (1125 morphed facial images and 1121 real facial images), validation set (571 morphed facial images and 564 real facial images), and test set (570 morphed facial images and 566 real facial images). In the HNU-FM dataset, two scenarios are considered. In scenario 1, the reference images utilize neutral expression images. In scenario 2, to simulate real-world scenarios, the reference images may include different expressions, makeup, glasses, etc. In both scenarios, only real images that are not used for morphing generation are selected as reference images.

The MIPGAN dataset utilizes the Face Research Lab London (FRLL) dataset [44] to generate morphed facial images, which includes MIPGAN-I and MIPGAN-II subsets. In the MIPGAN-I dataset, there are 837 morphed facial images, partitioned into training, validation, and test sets, comprising 503, 165, and 169 morphed facial images, respectively. Meanwhile, the MIPGAN-II dataset consists of 999 morphed facial images, also segmented into training, validation, and test sets, consisting of 599, 197, and 203 morphed facial images, respectively. Here, only the second scenario is considered for the MIPGAN dataset.

### 4.2. Implementation Details and Evaluation Metrics

For the facial images in the dataset, face alignment was conducted, and the facial images were cropped to a resolution of 256×256. Since the encoder and generator of this method both utilize pre-trained pSp [39] models and their weights are fixed, only the feature interaction-based de-morphing factor prediction network needs to be trained. The training of this network involves setting the epochs to 300, the batch size is set to 8, and using the Adam optimizer (with $\beta_{1}=0.9$, $\beta_{2}=0.999$) with a learning rate of ${1\times 10}^{-5}$ to adjust the model parameters. During the process of generating morphed facial images, a morphing factor between 0.1 and 0.9 is typically chosen for linear interpolation. Therefore, predicting the de-morphing factor can be framed as a multi-class classification problem. To optimize the feature interaction-based de-morphing factor prediction network, this method uses the cross-entropy loss function (CrossEntropyLoss). All experiments were conducted in a PyTorch 1.11 environment with CUDA version 12.1; the hardware configuration used was NVIDIA GeForce RTX 3090 24 GB GPU.

To evaluate the effectiveness of the proposed method, Face++ [43] is utilized to measure the similarity between the recovered facial image $I_{b}^{\prime }$ and the real facial images $I_{a}$ and $I_{b}$. The False Accept Rate (FAR) of Face++ is set to 0.1%, with a recommended threshold of 62.327. When the face recognition system determines that the recovered facial image $I_{b}^{\prime }$ matches the facial image of the accomplice $I_{b}$ but does not match the facial image of the criminal $I_{a}$, the restoration of the facial image is considered successful. The restoration accuracy is defined as:(6)$$Accuracy=\frac{T}{N},$$
where *N* represents the total number of restored facial images of accomplices, and *T* denotes the number of successfully restored facial images.

### 4.3. Ablation Experiments

#### 4.3.1. Effectiveness of the Feedforward Network

To evaluate the effectiveness of the feedforward network in the feature interaction-based de-morphing factor prediction network (DMFP), the feedforward network from the DMFP is removed, denoted as w/oFN. In this experiment, the backbone network used the pre-trained pSp model. The quantitative experimental results are shown in Table 1, and the comparison of recovery effects is illustrated in Figure 3.

The visual results in Figure 3 reveal that images restored by w/oFN are comparable to those restored by $DMFP_{pSp}$, but with some flaws. For instance, in the second example of Protocol I and the third example of Protocol III in Figure 3, facial distortion and unnatural color appearance are evident, respectively. As indicated in Table 1, the restoration accuracy of w/oFN decreases by approximately 2% compared to $DMFP_{pSp}$, and by 10% in Scenario 2 of Protocol III. This indicates that the absence of the feedforward network leads to poorer stability of the model when dealing with reference images containing different expressions, poses, makeup, and glasses. This underscores the importance of the feedforward network in extracting spatial and channel information, thereby enhancing the expression capability of the feature interaction-based de-morphing factor prediction network and facilitating the prediction of de-morphing factors.

#### 4.3.2. The Universality of the Feature Interaction-Based De-Morphing Factor Prediction Network

To assess the universality of the feature interaction-based de-morphing factor prediction network, this experiment compared models trained on the E2Style and pSp architectures for the StyleGAN face inversion, denoted as $DMFP_{E2Style}$ and $DMFP_{pSp}$, respectively. Their restoration accuracies are presented in Table 1. Additionally, the facial images restored by these models are illustrated in Figure 4.

From the restored images of $DMFP_{E2Style}$ and $DMFP_{pSp}$ in Figure 4, although some images restored by $DMFP_{E2Style}$ exhibit slightly unnatural colors in the faces (e.g., the fourth example in Protocol I in Figure 4), overall, it demonstrates restoration results similar to $DMFP_{pSp}$, generating facial images that are similar to the accomplice. As the quantified results in Table 1, $DMFP_{E2Style}$ also achieves good restoration accuracy. For the morphed faces with varying morphing factors, $DMFP_{E2Style}$ achieves restoration accuracies of 77.55% and 74.39% in Scenarios 1 and 2, respectively, which are close to those of $DMFP_{pSp}$. On the dataset with the morphing factor of 0.5, it achieves accuracies of 99.16% and 98.86%, slightly better than $DMFP_{pSp}$. Therefore, this indicates that the feature interaction-based de-morphing factor prediction network has good generalization capabilities for pre-trained StyleGAN-based face inversion networks.

### 4.4. Performance Comparison

To validate the performance of the proposed model, this experiment compared it with several existing methods: Face Demorphing [25] (utilizing a de-morphing parameter of 0.3 within the recommended range in [25]), FD-GAN [26], DAD [27], CNN [24], and cGAN [36]. The evaluation was conducted on both the HNU-FM dataset [41] and the MIPGAN dataset [42]. Notably, the DAD method [27], which restores facial images of two contributors from a single morphed facial image, does not differentiate between Scenario 1 and Scenario 2 on the HNU-FM dataset.

#### 4.4.1. Performance Comparison on HNU-FM Dataset

Comparative experiments were conducted on Protocol III of the HNU-FM dataset [41] to verify the effectiveness of the proposed method for landmark-based morphed facial images. The restoration accuracies of each method are presented in Table 2, and their visual results are illustrated in Figure 5.

The facial images restored by Face Demorphing [25] exhibit significant irregular color patches, particularly noticeable around areas such as the eyes and hair, as well as edge artifacts around the face, as seen in the first example of Scenario 2 in Figure 5. Furthermore, in Scenario 2 of Figure 5, both Face Demorphing [25] and cGAN [36] exhibit traces of eyeglass frames from the reference facial image. The facial images generated by DAD [27] appear closer to the morphed facial image, possibly due to its method of restoring facial images of two contributors from a single morphed image without relying on a reference image. While the facial images restored by CNN [24] and FD-GAN [26] show greater similarity to the facial images of the accomplice, they tend to be less clear and somewhat blurry. In comparison, the proposed method yields facial images with superior visual quality and closer resemblance to real accomplice facial images.

Quantitative results from Table 2 demonstrate that the proposed method surpasses existing face de-morphing techniques in restoration accuracy on the Protocol III dataset. In Scenario 1, the restoration accuracy of the proposed method is 80.71%, outperforming Face Demorphing [25], FD-GAN [26], CNN [24], cGAN [36], and DAD [27]. Notably, existing deep learning-based face de-morphing methods experience a significant decrease in recovery accuracy in Scenario 2. However, the proposed method maintains a restoration accuracy of 79.48% in Scenario 2, representing only a marginal decrease of 1.23% compared to Scenario 1. This indicates the better stability of the proposed method in scenarios where the reference image may contain complex elements such as expressions and glasses.

To evaluate the performance of the proposed method for morphing face images with a morphing factor of 0.5, models trained on Protocol III of the HNU-FM dataset are generalized to Protocol I for a comparative experiment. Table 3 presents the recovery accuracy of all methods on Protocol I, while their visual results are displayed in Figure 6.

The restoration results depicted in Figure 6 reveal challenges faced by Face Demorphing [25], DAD [27], cGAN [36], and FD-GAN [26] when restoring images on Protocol I, similar to those observed on Protocol III. Furthermore, the CNN model [24] produces facial images that appear relatively darker on Protocol I, with a poorer restoration effect observed in the eyes. In contrast, the proposed method outperforms other methods on Protocol I, exhibiting superior visual quality and similarity in the restoration results.

The quantitative results in Table 3 reveal that, compared to Protocol III, most face de-morphing methods exhibit improved restoration accuracy on Protocol I. However, there is a noticeable decrease in performance for the DAD model [27] on this dataset, possibly due to the recovered facial images closely resembling the morphed face images. Notably, the proposed method achieves a restoration accuracy of 98.31% in Scenario 1 and 92.23% in Scenario 2, surpassing existing methods.

#### 4.4.2. Performance Comparison on MIPGAN Dataset

Comparative experiments are also conducted on the MIPGAN dataset [42] to evaluate the proposed method’s capability to restore morphed faces generated by GAN. Table 4 presents the restoration accuracy of all methods on this dataset, while their visual results are illustrated in Figure 7.

The restoration results depicted in Figure 7 reveal that, for the MIPGAN dataset [42], Face Demorphing [25] encounters similar challenges observed on the HNU-FM dataset [41]. Furthermore, the facial images reconstructed by Face Demorphing [25] exhibit distortions in the mouth area, as evidenced in the fifth example in Figure 7. This might be due to the different distribution of facial landmarks between the morphed facial image and the real faces, posing challenges for landmark-based restoration. The facial images restored by the DAD [27] and cGAN [36] models still resemble the morphed facial images on the MIPGAN dataset, as demonstrated by the second example in Figure 7. Additionally, the facial images restored by the DAD model [27] exhibit some noise. Although the facial images restored by the CNN [24] and FD-GAN [26] are similar to the accomplices’ faces, they suffer from blurriness and exhibit some edge artifacts. Compared to other methods, the proposed method successfully recovers facial images similar to those of the accomplices on the MIPGAN dataset and demonstrates high visual quality.

The restoration results in Table 4 highlight the superior performance of the proposed method on the MIPGAN dataset. On the MIPGAN-I dataset, the proposed method achieves an impressive restoration accuracy of 85.80%, outperforming Face Demorphing [25], the CNN model [24], the DAD model [27], the cGAN model [36]%, and FD-GAN [26]. Similarly, on the MIPGAN-II dataset, the proposed method achieves a remarkable restoration accuracy of 91.38%, significantly surpassing existing methods.

#### 4.4.3. Analysis of Restoration Effects

To evaluate the effectiveness of the proposed method in facial restoration, we measured the average matching scores between the restored facial images and those of both the accomplice and the criminals using Face++ [43]. The evaluation adhered to the recommended threshold score of 62.327. For successful reconstruction of the accomplice’s facial image, the restored image must closely resemble the accomplice’s appearance while maintaining distinction from the criminal’s image. Therefore, a higher average matching score between the restored facial image and the accomplice indicates better performance, whereas a lower score with the criminal indicates superior performance. Moreover, a greater disparity between these average scores signifies a more effective restoration. These results are depicted in Figure 8, where the upper edge of each rectangle represents the average matching score between the restored image and the accomplice’s facial image and the lower edge represents the score between the restored image and the criminal’s facial image.

The results depicted in Figure 8 reveal that, across both the HNU-FM [41] and MIPGAN datasets [42], $DMFP_{pSp}$ effectively balanced the matching scores between the restored facial images and the accomplices and the criminals. The restored facial images exhibited high similarity to the accomplice while maintaining differentiation from the criminal. In contrast, the DAD model [27] produced facial images with average matching scores between the accomplice and the criminal that exceeded the recommended threshold, with a relatively small disparity between the two scores. Additionally, although the facial images restored by Face Demorphing [25] were similar to the accomplice, they lacked clear differentiation from the criminal. While cGAN [36] achieved a reasonable balance between the two average matching scores, the score between the restored facial image and the accomplice’s facial image approached the threshold, indicating a lesser preservation of the accomplice’s identity features. Furthermore, both the CNN model [24] and FD-GAN [26] achieved a satisfactory balance, but the distance between the two average matching scores was shorter, indicating a lesser restoration effect compared to $DMFP_{pSp}$. The experiments demonstrate that the proposed method produces facial images with superior restoration effect.

#### 4.4.4. Evaluation of Model Performance

The proposed model is quantitatively compared with Face Demorphing [25], FD-GAN [26], DAD [27], CNN [24], and cGAN [36] in terms of both model capacity and inference time. The results are presented in Table 5. Although the proposed method $DMFP_{pSp}$ has more parameters than the other methods, only the de-morphing factor prediction network (DMFP) is trained, which has fewer parameters than FD-GAN [26] and DAD [27], making it easier to train. Additionally, the use of a pre-trained StyleGAN enhances the effectiveness of the face de-morphing task, leading to high-quality restoration of accomplice facial images. In terms of inference time, $DMFP_{pSp}$ achieves the best performance, processing 15.58 images per second, which can meet the needs of real-time forensics.

## 5. Conclusions

The experimental results demonstrate that the proposed method, evaluated on both the HNU-FM and MIPGAN datasets, outperforms previous approaches in terms of restoration accuracy and image quality. Additionally, the feature interaction-based de-morphing factor prediction network shows strong applicability when integrated with pre-trained StyleGAN face inversion models. This highlights the potential for restoring the accomplice’s facial image by predicting the de-morphing factor and utilizing pre-trained StyleGAN. However, the use of inverse linear interpolation in the latent space to obtain the identity features of accomplices, after predicting the de-morphing factor, is simplistic. In the future, we will focus on developing more sophisticated methods to accurately capture the identity features of the accomplice.

## Figures and Tables

**Figure 1 sensors-24-05504-f001:**
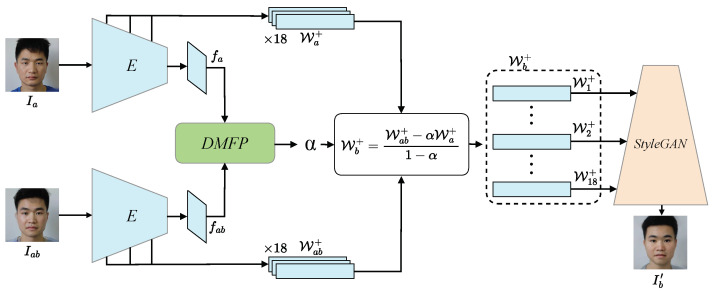
The overall architecture of the proposed method. It consists of a pre-trained StyleGAN-based facial inversion model and a feature interaction-based de-morphing factor prediction network.

**Figure 2 sensors-24-05504-f002:**
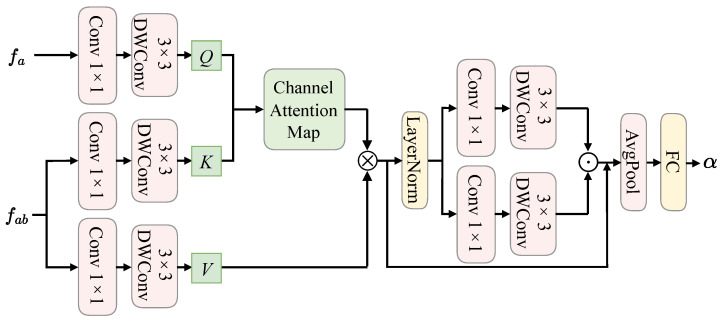
The structure details of the feature interaction-based de-morphing factor prediction network.

**Figure 3 sensors-24-05504-f003:**
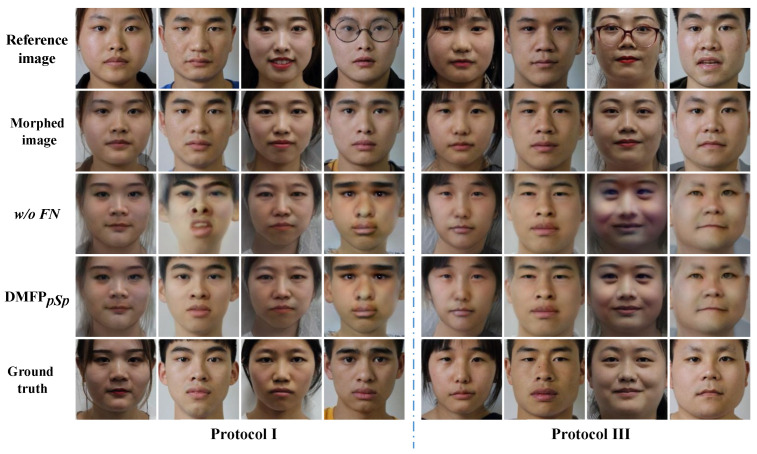
Visual comparison of different variant models. In Protocols I and III, the first two columns show the recovery results for Scenario 1, while the last two columns exhibit the recovery results for Scenario 2.

**Figure 4 sensors-24-05504-f004:**
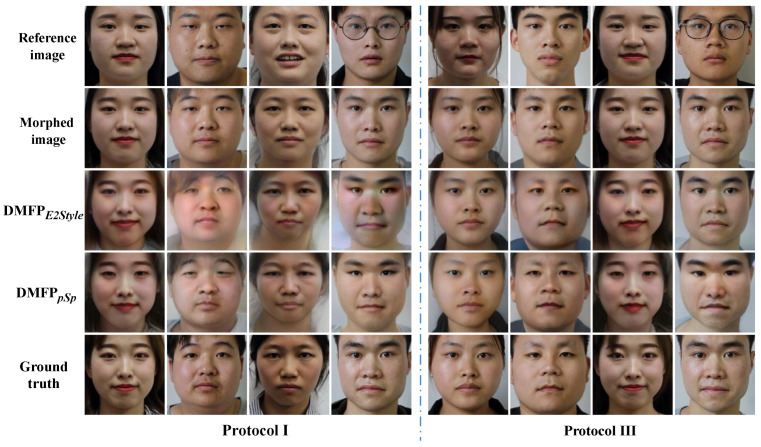
Visual comparison of different pre-trained StyleGAN models. In Protocols I and III, the first two columns show the recovery results for Scenario 1, while the last two columns exhibit the recovery results for Scenario 2.

**Figure 5 sensors-24-05504-f005:**
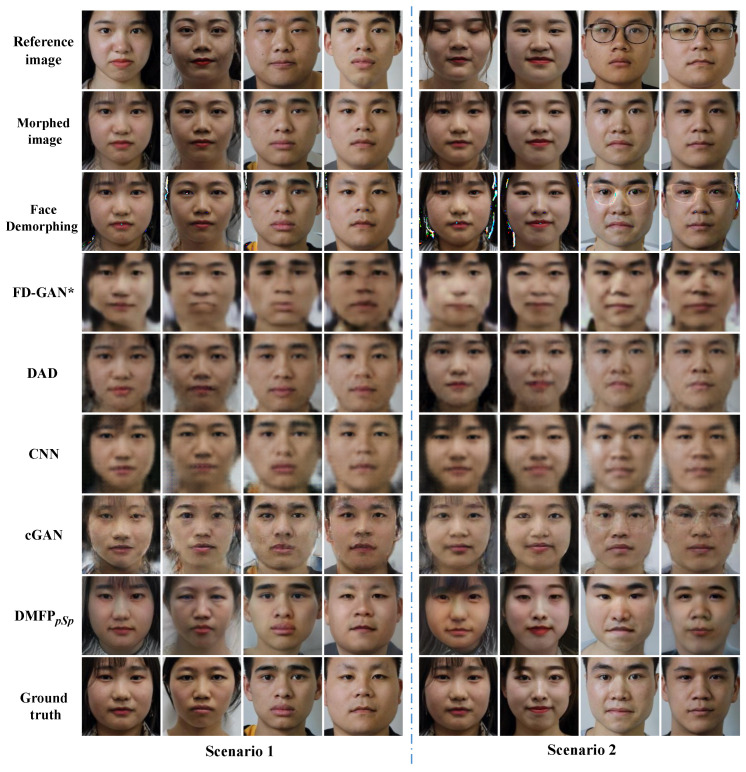
Visualization results on Protocol III of the HNU-FM dataset (* indicates that the model is reproduced in the Pytorch environment).

**Figure 6 sensors-24-05504-f006:**
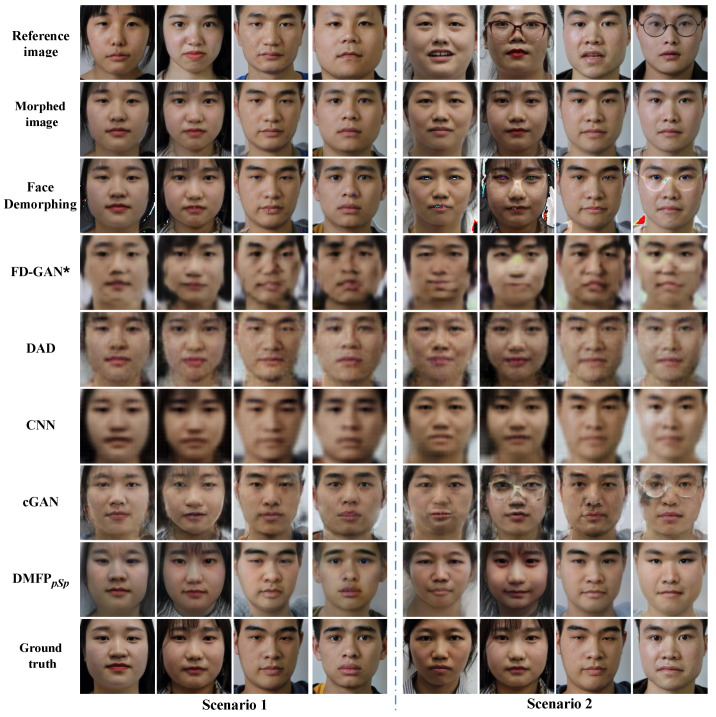
Visualization results on Protocol I of the HNU-FM dataset (* indicates that the model is reproduced in the Pytorch environment).

**Figure 7 sensors-24-05504-f007:**
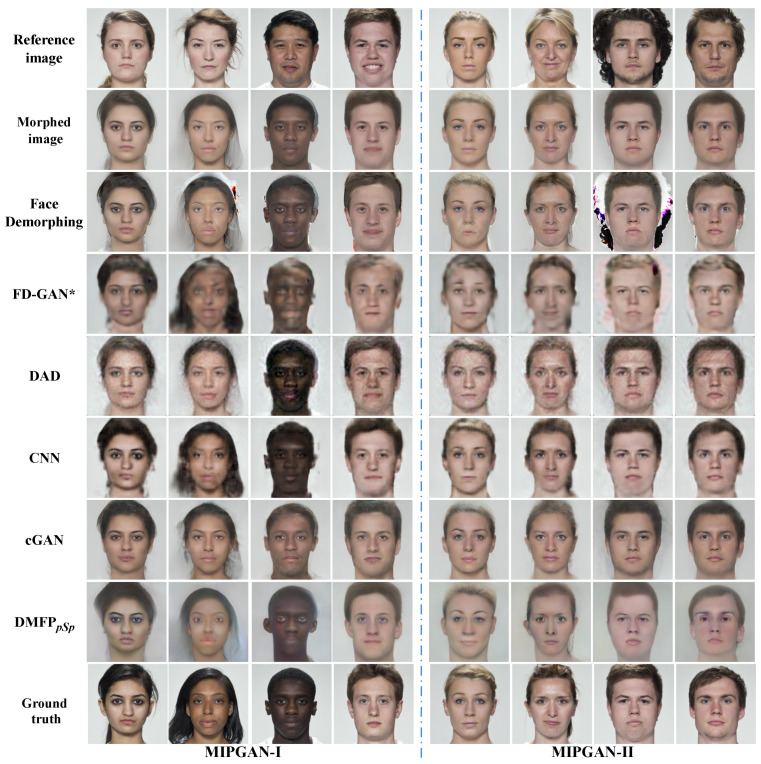
Visualization results on the MIPGAN dataset (* indicates that the model is reproduced in the Pytorch environment).

**Figure 8 sensors-24-05504-f008:**
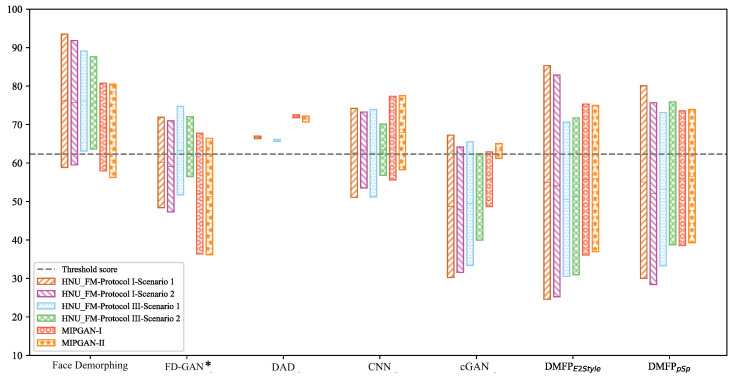
The average matching scores between restored facial images and those of both accomplices and criminals. Each rectangle’s upper edge denotes the average matching score between restored facial images and those of accomplices, while the lower edge represents the average matching score with criminals’ images (* indicates that the model is reproduced in the Pytorch environment).

**Table 1 sensors-24-05504-t001:** The restoration accuracy with ablation on the HNU-FM dataset.

Method	Protocol I		Protocal III	
	Scenario 1	Scenario 2	Scenario 1	Scenario 2
w/oFN	96.12%	90.03%	78.42%	69.48%
$DMFP_{E2style}$	**99.16%**	**98.86%**	77.55%	74.39%
$DMFP_{pSp}$	98.31%	92.23%	**80.71%**	**79.48%**

Bold indicates the best results.

**Table 2 sensors-24-05504-t002:** Restoration accuracy on Protocol III of the HNU-FM dataset.

Method	Face Demorphing [25]	FD-GAN [26] *	DAD [27]	CNN [24]	cGAN [36]	${\mathrm{DMFP}}_{\mathrm{pSp}}$
Scenario 1	49.82%	66.49%	44.21%	65.79%	64.91%	**80.71%**
Scenario 2	49.12%	53.73%	44.21	54.39%	53.33%	**79.84%**

Bold indicates the best results. * indicates that the model is reproduced in the Pytorch environment.

**Table 3 sensors-24-05504-t003:** Restoration accuracy on Protocol I of the HNU-FM dataset.

Method	Face Demorphing [25]	FD-GAN [26] *	DAD [27]	CNN [24]	cGAN [36]	${\mathrm{DMFP}}_{\mathrm{pSp}}$
Scenario 1	56.08%	77.03%	36.49%	77.03%	75.34%	**98.31%**
Scenario 2	52.70%	70.61%	36.49	67.91%	63.51%	**92.23%**

Bold indicates the best results. * indicates that the model is reproduced in the Pytorch environment.

**Table 4 sensors-24-05504-t004:** Restoration accuracy on the MIPGAN dataset.

Method	Face Demorphing [25]	FD-GAN [26] *	DAD [27]	CNN [24]	cGAN [36]	${\mathrm{DMFP}}_{\mathrm{pSp}}$
MIPGAN-I	56.21%	75.74%	24.85%	69.82%	50.89%	**85.80%**
MIPGAN-II	66.01%	71.43%	25.12%	60.01%	32.02%	**91.38%**

Bold indicates the best results. * indicates that the model is reproduced in the Pytorch environment.

**Table 5 sensors-24-05504-t005:** Quantitative comparison of model performance.

Method	Model Capacity	Inference Time
	(Million)	(img/sec NVIDIA 3090 24 GB)
Face Demorphing [25]	/	0.55
FD-GAN [26] *	44.00	10.50
DNN [27]	43.80	6.50
CNN [24]	4.16	5.21
cGAN [36]	54.53	13.04
DMFP	9.12	/
$DMFP_{pSp}$	252.53	15.58

* indicates that the model is reproduced in the Pytorch environment.

## Data Availability

All the data used are contained in the paper.

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
