# Peer review of "Feature Interaction-Based Face De-Morphing Factor Prediction for Restoring Accomplice’s Facial Image"

_sensors, 2024, doi:10.3390/s24175504_

Round 1
Reviewer 1 Report
Comments and Suggestions for Authors
I recommend that the authors take into consideration the following comments:
1."Fig. 2" should be removed from Figure 2; please confirm whether the illustrations in Figures 3 and 4 are for Protocol III.
2.Please check if the text following the equations needs to be indented, especially in lines 230, 258, and 343, and ensure consistency in capitalization, such as in lines 268 and 318. This requirement applies to all equations.
3.It is necessary to emphasize novelty and contributions, clearly specifying the improvements to highlight the main differences from previously published works.
4.The introduction and related work sections consist of several short paragraphs. It is recommended to reorganize this part to improve readability.
5.The innovation of the article is not evident. The channel attention mechanism and StyleGAN method used have not been improved before their application. There is little theoretical analysis on the selection of these methods, and the comparison is lacking.
6.What are the limitations of this method? Please briefly state them in the conclusion section.
7.There are issues with the authors' names in the references, such as in lines 555, 557, and 563. Please check all references.
s, such as in lines 555, 557, and 563. Please check all references.
Comments on the Quality of English LanguageMinor editing of English language is required.
Reviewer 2 Report
Comments and Suggestions for Authors
The manuscript is structured and well written, clearly describes the problem statement, gives an overview of face de-morphing methods, describes the motivation and novelty of the proposed method.
A few comments:
1. I found a similar paper by some of the same authors:
Long, M., Zhou, J., Zhang, L.-B., Peng, F., Zhang, D.: ADFF: Adaptive De-Morphing Factor Framework for Accomplice Face Image Restoration. IET Image Process. 18, 470-480 (2024). https://doi.org/10.1049/ipr2.12962
There seems to be no reference to this paper in the manuscript. The question is about the relationship between this paper and the manuscript. Also, Table 2 from the manuscript is very similar to Table 4 from the paper, but the numbers are different. Please explain this.
2. Do the authors have their implementation in the public domain so that the reproducibility of the results from the manuscript is possible?
3. Please comment on the computational cost of the proposed method.
4. Can the authors add some information about the method and the results, e.g. the predicted de-morphing factors?
5. More refined results on accuracy, e.g. separately for men/women, influence of glass and hair, etc. would be helpful to understand the sources of the advantage of the proposed methods.
Round 2
Reviewer 1 Report
Comments and Suggestions for Authors
Please avoid to write in first person, i.e., replace phrases like "... We propose a novel face de-morphing method.." with "... this work We propose a novel face de-morphing method..". This comment applies for all section of the wholemanuscript.。
Comments on the Quality of English LanguageMinor editing of English language required.
Reviewer 2 Report
Comments and Suggestions for Authors
The authors responded to most of my comments. Some of them may be answered in future research.
Typos: In a couple of added parts of sentences after formulas, lowercase letters should be after commas.
